# Stunning Compliance in Halal Slaughter: A Review of Current Scientific Knowledge

**DOI:** 10.3390/ani13193061

**Published:** 2023-09-29

**Authors:** Awis Qurni Sazili, Pavan Kumar, Muhammad Nizam Hayat

**Affiliations:** 1Halal Products Research Institute, Universiti Putra Malaysia (UPM), Putra Infoport, Serdang 43400, Malaysia; 2Department of Animal Science, Faculty of Agriculture, Universiti Putra Malaysia (UPM), Serdang 43400, Malaysia; nizamhayat88@yahoo.com.my; 3Institute of Tropical Agriculture and Food Security, Universiti Putra Malaysia (UPM), Serdang 43400, Malaysia; pavankumar@gadvasu.in; 4Department of Livestock Products Technology, College of Veterinary Science, Guru Angad Dev Veterinary and Animal Sciences University, Ludhiana 141004, India

**Keywords:** religious slaughter, halal, stunning, unconsciousness, animal welfare

## Abstract

**Simple Summary:**

Stunning renders animals to an unconscious state before slaughter improves animal welfare by reducing stress, alleviating pain, and minimizing fear. Muslim scholars are not unanimous on the issue of the application of stunning in the halal slaughtering of animals as some scholars perceived it as more painful, obstructing free blood flow, and causing the death of animals prior to the halal cut. Scientific findings suggest that halal compliance stunning technologies are reversible, do not kill animals prior to the halal cut, and do not obstruct blood loss. The present review summarizes various stunning methods and their suitability for application in the halal slaughter.

**Abstract:**

Muslim scholars are not unanimous on the issue of the application of stunning in the halal slaughtering of animals. Appropriate stunning makes animals unconscious instantaneously, thus avoiding unnecessary pain and stress during the slaughtering of animals. The present review comprehensively summarizes the available scientific literature on stunning methods in view of their halal compliance during the slaughter of animals. The issue of maximum blood loss, reversibility of consciousness, and animals remaining alive during the halal cut are the key determinants of approval of stunning in the halal slaughter. Further, missed stuns due to poor maintenance of equipment, improper applications, and poor restraining necessitates additional stunning attempts, which further aggravates pain and stress in animals. Scientific findings suggest that halal-compliant stunning technologies are reversible, do not kill animals prior to the halal cut, and do not obstruct blood loss. There is a need to carry out further research on the refinement of available stunning technologies and their application, proper restraints, proper identification of the death status of animals, and assurance of animal welfare in commercial halal meat production.

## 1. Introduction

Religious slaughter remains a contentious and emotive issue, encompassing animal welfare issues, human rights, freedom of religion, consumer rights, and market power [1]. Usually, the slaughtering procedure is performed by severing the main blood vessels (carotid arteries that supply oxygenated blood to the brain and jugular veins that carry deoxygenated blood from the brain and other cells back to the heart). Death is attained due to the lack of oxygen supply to the brain [2]. The process, in general, would induce pain and stress to the animal, which, if not carefully monitored, would compromise meat quality. The pain sensation and stress suffered by the animal would jeopardize the animal welfare aspects, which is the primary concern for both the Muslim community and Western countries [3,4,5]. 

Pain refers to a “discriminative sensation comprising emotional experience caused by actual or potential tissue damage” [6]. The perception of pain is protective and warns animals against situations that may damage tissues and avoid these situations in the future [7]. A noxious stimulus is a stimulus that causes or may cause tissue damage, thereby causing pain perception. It stimulates nociceptors (physiological receptors/sensory receptors) that generate electric impulses in the associated nerves, later carrying the information to higher centers for further processing as pain. The pre-sensitization of nociceptors decreases their threshold and, consequently, perception of pain to stimulation that usually does not induce pain perception (allodynia). Nociceptors are activated during improper preslaughter handling and slaughter, such as neck cuts, injury, sticking, electric prods, and a sharp knife, thereby generating electric impulses that are transmitted along the neuronal axon to the spinal cord–brain stem–thalamus–cortex for further processing as pain [8,9]. The prolonged activation of nociceptors due to improper stunning can cause hyperalgesia and peripheral sensitization (increase in the sensitivity of afferent neurons to stimulations) or central sensitization due to the release of neuropeptides and upregulating the existing receptors [10,11]. The central and peripheral sensitization before slaughter also increases the nociception perception during slaughter [11,12,13]. During animal slaughter without stunning, the cutting of soft tissues (muscle, tissue, and viscera) of the neck induces the nociceptive nerve fibers to generate electric impulses, which are transmitted to the higher center of the nervous center for its interpretation as pain [14,15].

In order to overcome these issues, various stunning procedures have been developed in order to eliminate pain during slaughter. Stunning makes animals unconscious instantaneously and is widely used in the slaughter of animals throughout the world, even made compulsory during slaughter in some countries. With the increasing Muslim population, increase in net income, education, awareness, and meat quality assurance, the demand for halal meat is rapidly increasing throughout the world. However, Muslim scholars are not unanimous on the application of stunning during halal slaughter. The present review summarizes various aspects of the application of stunning technologies during the halal slaughtering of livestock. 

## 2. Halal Slaughter

Halal is an Arabic word associated with the Muslim community and the Islamic religion. The word “halal” can be translated to lawful, permissible, valid, approved, sanctioned, legal, authorized, or trustworthy [16]. The term halal is stated in several chapters in the Holy Qur’an, and also in Hadiths, the saying of Prophet Muhammad (PBUH). Although the term halal, in most cases, is associated with the type of meat, food, and ingredients that are acceptable/permissible to be consumed by Muslims, in reality, the word halal carries a more extensive definition that includes a whole different aspect such as the preparation, the source, and condition in which food is produced [17]. For instance, meat obtained from a slaughtering process (*Zabiha*/*Dhabihah*) needs to fulfill several criteria before it can be considered as halal meat. The animal intended to be slaughtered must be deemed alive (*Alhayat Almustaqirah*) and sound, and God’s name should be evoked prior to the incision cut (Qur’an 6: 121). Failure to abide by the aforementioned rules will cause the meat to be considered as non-halal, or unfit for consumption by Muslims, as mentioned in surah Al-An’am 6: 119.

“And why should you not eat of that meat on which Allah’s Name has been pronounced (at the time of slaughtering the animal), while He has explained to you in detail what is forbidden to you…” (Al-An’âm 6: 119)

The current practice of halal slaughter is derived from the Shariah law interpreted from the Qur’an and Hadith. As per Shariah law, *Zabiha/Dhabihah* must meet the following conditions such as: Animal must be alive at the point of slaughter (Qur’an 5: 3, 6: 118–119, 6: 145, 16: 115).Recitation of the Holy name (*Tasmiyyah*) Allah during slaughter (Qur’an, 6: 118–119, 22: 34, 22: 36).“*and do not eat from animals on which the name of Allah is not mentioned*” (Qur’an 6: 121)The slaughterman has attained the age of discretion and is mentally stable. Slaughtering by a Muslim is preferred. However, Muslims can consume meat slaughtered by the people of the Book (Qur’an 5: 5).

The followings are some recommendations and should be encouraged to follow these during slaughtering as: Orientation of the animal to face Qibla.Sharpen the blade or knife away from the sight of the animal.Slaughtering animals out of sight of other animals and shielding animals from the sight of blood.Slaughtering in one single movement of the knife.

Riaz et al. [18] summarized the basic requirement of halal slaughter as per the Gulf Standard Organization (GSO 993/1998) as follows: The animal must be alive during slaughtering, and any pre-slaughter handling should not cause death to the animal.Stunning should be reversible, with animals gaining full consciousness if not slaughtered.Recitation of Allah at the time of the slaughtering of each animal by the slaughterman.Use of a very sharp knife and making the cut with one continuous strike.The slaughtering process should be done from the anterior to the neck end.No cutting of the head during slaughtering to facilitate proper bleeding.Other processing or handling operations must be performed after the death of an animal without an eye reflex.Maximum blood should be drained out of the carcass, resulting in animal death due to cerebral anoxia in the absence of circulatory blood.

## 3. Stunning

There is a debate on the state of the animal during slaughtering; for example, some Islamic scholars believe that animals should be alive during slaughtering, while other Islamic scholars conclude that animals should be conscious during slaughter [19]. This is due to the two different interpretations of “animal must be alive during slaughter” stipulated in the Holy Qur’an. Some Islamic scholars interpreted it as animals should be conscious, whereas others interpreted the same as pumping the heart will suffice [20]. The Shafi’i and Hanbali scholars postulated the three states or levels of life in the context of Halal slaughter, [21] as:i.Hayah al-Mustamirrah: Normal and ordinary life whereby the lifespan of an animal ends with its slaughter or due to other factors.ii.Hayah al-Mustaqirrah: An animal has a strong will to live, shows signs of life, and has the ability to move. The spray of blood after severing the windpipe and veins indicates the sign of life.iii.The state in which an animal has signs of uncontrolled movement and lost its ability to see and hear [22].

Basically, consciousness consists of two components: content of consciousness and level of consciousness. The content of consciousness refers to the awareness of the environment and inner states, which needs proper functioning of the cerebral cortex, connecting networks to subcortical structures. The level of consciousness refers to wakefulness [23], regulated by the ascending reticular system in the upper brainstem tegmentum and thalamus. In the slaughter context, perception of the environment, other animals, and visual threats are important determinants. However, there is a need for in-depth studies to fully understand their roles in the context of pain perception during slaughter. In addition, pain perception is considered a “conscious sensation” that warrants a functioning cerebral cortex [24]. In addition, the recent evidence of low levels of residual pain perception during an unconscious state makes this issue very complex [25]. 

In general, according to Terlouw et al. [26] consciousness is the state when the signs of consciousness (viz., standing posture, head and body righting reflexes, voluntary vocalization, response to painful stimuli, eye movements, and natural blinking) are present, and signs of unconsciousness are absent (viz., absence of corneal, eyelash, and rhythmic reflexes). The presence of rhythmic breathing immediately after stunning and other signs of vocalization and spontaneous blinking are signs of ineffective stunning [27,28,29,30,31]. Grandin et al. [11] summarized various vital indicators of consciousness in buffalo as attempts to regain posture and collapse failure, head raising, eyelid and corneal reflexes with blinking and full eyeball rotation, and respiratory rhythm. 

In addition, during slaughter, animals are more prone to stress and injuries such as transport injuries, disease conditions, injuries during handling, trauma caused by painful lesions due to sticks or tubes, and lack of habituation to humans. This aggravates the pain perception by sensitization of neurons and enhances transmission and pain perception [12]. The other factors that may cause pain during slaughter are methods of restraints, the sticking process [32] and aspiration of blood in the lungs [33]. 

Various signs for evaluating unconsciousness during the slaughter of water buffalo are summarized in Figure 1.

Stunning renders animals to an unconscious state before slaughter. It is acclaimed to relieve the distress and pain of animals before slaughter. Still, there is a lack of consensus on applying stunning in halal slaughter. Traditionally, halal slaughtering is performed without stunning. As stunning comes under practice after revealing the Qur’an, some scholars thus considered this as haram (unlawful) as it was not practiced by Prophet Muhammad (PBUH) nor described in the scriptures. In addition, the stunning of animals has not been described as haram, and the Shariah law and Hadith did not have a clear message as such due to its invention at a later phase. For applying any new technologies not described in scripture or discovered later, Islam has a set of mechanisms of their acceptance/adaptation based on the fatwa (judgment) issued by Islamic scholars. Further, Ijma’ (consensus on legal opinion) as well as Qiyas (reasoning by analogy) to suit the place, time, and situations have an important place in Islam. However, Muslim scholars have started approving stunning to fulfill the legal requirements of slaughter [34]. In some cases, such as the application of stunning during the slaughter of animals, Islamic scholars differ in their view and have different interpretations of Shariah law.

### 3.1. Opponent of Stunning

Fuseini et al. [35] summarized the following factors that affect the acceptability of stunning in halal slaughtering such as:Death of an animal is caused by pre-slaughter stunning and not due to ritual neck cut and exsanguination [36].The pre-slaughter stunning and mechanical slaughter technologies have not been mentioned in the Holy Qur’an [36].New technology may decrease blood loss volume from the carcass [37].Fear of not severing carotid arteries and jugular veins on both sides.Pre-slaughter stunning also causes pain and suffering and is inhumane [35,38].Negative impact of stunning on meat quality.

As per the Qur’an (Qur’an, verse 5.3), it is forbidden to eat meat from animals killed by the blow. Some Islamic scholars interpret that the Qur’an does not permit any form of mechanical stunning. There are elements of doubt regarding the animal that would remain alive during slaughter after stunning; thus, meat from stunned animals is not preferred by several consumers. The main concern during the slaughtering of stunned animals is their possibility of death even before neck cuts. Lever and Miele [39] and the Halal Monitoring Committee (HMC) [40] observed that even after the voltage and size of birds were standardized, some birds were still dead due to stunning before exsanguination, mainly attributed to the delayed stun to sticking time. A higher electric current could stop the heart beating and increase red wingtips [41]. 

The Humane Methods of Slaughter Act (1958), enforced by USDA Food Safety and Inspection Services (FSIS) and European Council Regulation, EC 1099/2009, allows slaughter without stunning for religious purposes. The meat produced without stunning is still regarded by most Muslims as having the highest spiritual quality as the same was earlier practiced by Prophet Muhammad (PBUH) and the earlier Biblical prophets [40,42,43]. Lower bleed-out, poor meat quality, carcass quality, and gross animal welfare violations are commonly encountered due to improper stunning. Some Islamic scholars have argued that lower blood loss in stunned animals is due to changes in vascular, neurological, and cardiovascular systems [44] and some claim this from religious scriptures [36]. There is a perception among some scholars and researchers that stunning causes more pain to animals than slaughtering [19,45,46]. However, this could be primarily due to the chances of mis-stun and associated hyperalgesia leading to compromised animal welfare and intense pain perception [7]. 

There is a severe compromise on animal welfare issues if animals are mis-stunned or improperly stunned. However, with the improvement of stunning technologies, continuous monitoring, and stringent legal framework, the incidences of mis-stun are markedly reduced. Proper restraint is also crucial in reducing stress during the slaughtering process. Bager et al. [47] and Grandin and Regenstein [48] also noted very little or no pain in calves and bulls during throat cuts without stunning in an upright restraint system. Animals stood still without any movement except a shudder noticed when the blade touched the throat, and it was noted as less vigorous than an animal’s response upon putting an ear tag. Authors noted that during throat cuts, “animals were not aware that their throat had been cut”. Gibson et al. [49] recommended a high neck cut during the slaughter of cattle to ensure animal welfare as indicated by a significantly reduced final collapse time in high neck cut at first cervical vertebrae (12–15 s) as compared to 17–20 s for final collapse taken by cattle upon traditional halal neck cut at the second to third cervical vertebrae. This could be due to higher branches of carotid arteries at the C1 vertebra, leading to rapid blood loss and fewer incidences of false aneurysms. 

### 3.2. Stunning Methods in Halal Slaughtering

As per Malaysian Protocol for Halal meat and poultry production, the Shariah requirements of stunning, as summarized by Nakyinsige et al. [50], are as follows: Stunning must be reversible and should not cause permanent injury and death to the animal.The operator of stunning equipment should be preferably a Muslim and be adequately trained in its application.Proper verification of halal compliance of stunning by a Muslim halal checker.Animal should be alive or deemed to be alive at the time of slaughter.Animals that died due to stunning should be removed from slaughtering.Invoking the phrase “Bismillah Allahu Akbar” by a Muslim slaughterman immediately before halal cut.Complete and spontaneous bleed-out.Dressing of ruminants and scalding of poultry only start after the death of the animal due to bleeding.Stunning equipment used for stunning halal animals should not be used for stunning animals considered haram by Shariah law.If stunning equipment was previously used for stunning haram animals, then it should be ritually cleansed under the supervision and verification of a competent Islamic authority.A separate provision of premises for stunning halal animals.

Pre-slaughter stunning is becoming widespread during halal slaughter as many Muslim-majority countries have approved such practice. However, the likelihood of death of animals prior to slaughter due to stunning and removal of such animals before halal throat cut remain major concerns. There is an increasing concern among Muslim consumers towards proper halal compliance during the stunning of animals due to potential violation of fundamental principles mentioned in the scriptures, such as animals must remain alive during slaughter and proper animal welfare compliance [50]. Only stunning technologies in which the animal is not dead before slaughter and the stunning process is reversible are recommended for halal slaughtering and accepted in halal slaughtering of animals [51]. In addition, some religious authorities accepted stunning in halal slaughter provided the animal’s heart is beating during sticking and bleeding [52,53].

Islam is a comprehensive religion and has emphasized animal welfare protection and proper handling of animals, and it forbids any harm or pain to animals throughout the animal’s life. In this context, some Islamic scholars advocate that stunning makes animals unconscious and does not cause death before slaughter, thus, stunning before slaughter is halal and acceptable [54]. Further, this technology is not explicitly prohibited in the Qur’an and Hadith, thus as long as the other commandments are followed (such as animals not dead during slaughter, as per Qur’an 2: 173, 5: 3), stunning is permitted. However, before approving a stunning technology for commercial halal meat production, it should be appropriately verified for reversibility and not cause permanent animal injury. Various behavioral and physiological reflexes are absent in the effectively stunned animals due to the damage caused to the cerebral hemispheres, thalamic structures, or the ascending reticular activation systems [11,55].

Stunning has been mandatory for slaughtering animals in the European Union since 1979 with a provision for member states to grant exemptions for religious slaughter. Denmark, Sweden, Slovenia, Iceland, and Norway make it compulsory to stun animals before slaughter, even for religious slaughter, whereas Lichtenstein and Switzerland mandated prior stunning before slaughter except for poultry. In addition, Austria, Estonia, Greece, and Latvia also mandate post-cut stunning [56]. In New Zealand, there is no exemption for stunning requirements given for religious slaughter. Germany, the United Kingdom, Australia, and Poland allow exemptions from preslaughter stunning for religious slaughter. However, there are reports on applying stunning methods that potentially lead to reversible stunning, such as captive bolt stunning, applied during the halal slaughter of animals [19,45,53]. In the irreversible stunning, there may be chances of animals dying even before the neck cut, thus violating the original concept of the animal remaining alive at the time of the neck cut. There should be a minimum time lag between stunning and sticking. In addition, there is a common perception among the public that animals without stunning bleed out better than stunned animals [54]. 

With the increasing focus on the application of stunning in slaughter due to its role in alleviating unnecessary pain/suffering, this technology is widely approved by several Muslim-majority countries such as Malaysia, Saudi Arabia, Indonesia, the United Arab Emirates, Egypt, and Yemen, for halal slaughtering of animals. Halal Food Authority (HFA), London, UK advocated stunning to make animals unconscious followed by neck cut during slaughter.

Some significant milestones in this approving stunning of animals in halal slaughtering are as follows:1978: Fatwa issued by the Egyptian Fatwa Council at Al Azhar University regarding the suitability of electro-narcosis for halal slaughter.1987: Fiqh Council in Makkah, Saudi Arabia in 10th Islamic Fiqh Council at the Muslim World League regarding the application of reversible electrical stunning.2006: Council for Legal Verdicts in Yemen regarding reversible electrical stunning.

During slaughter without stunning, the time gap between the throat cut and loss of consciousness of animals should be kept as minimal as possible to minimize the duration of potential pain and suffering by animals. Further, an extended time to lose consciousness with minimum pain and distress is more ethical than a shorter duration of more pain and suffering (Regenstein, personal communication cited by Khalid et al. [57]). An effective neck cut in lamb induces unconsciousness within 2–7 s and cortical brain death within approximately 14 s [58]. Rodriguez et al. [59] observed that in case of inefficient bleeding, onset of unconsciousness in lambs could be extended to 60 s. To prevent regaining consciousness, Grandin [60,61] advocated the stun to neck cut time <20 s for goats, 12 s for calves, and 23 s for cattle. 

There are concerns that blood aspiration into the upper respiratory tract and lungs of animals during slaughter without stunning causes suffering [62], whereas some hold the view that there will be no suffering because afferent signals activated by lung irritants are conveyed by neurons in the vagus nerves [63], and these are severed during slaughter without stunning. Gregory et al. [64] observed the aspiration of blood in the respiratory tract of cattle during traditional halal slaughter without stunning, with 58% of slaughtered cattle having inner blood lining in the trachea and 69% of slaughtered cattle having inner blood lining in the upper bronchi. From an animal welfare point of view, pain from the point of slaughter cut to unconsciousness, there may be a chance of blood aspiration into the lungs and irritation in airways as breathing continues in non-stunned animals [15,64]. 

In non-stunned animals, false aneurysms in carotid arteries at cardiac and cephalic ends caused by retracting its end within its surrounding connective tissue sheath caused prolonged consciousness (more than 60 s in cattle as compared to 40–60 s in normal situations) and improper bleeding out of the animal [33,65,66]. During bleeding, the adventitia comes in contact with blood and becomes swollen, which may impede the blood flow by sealing the severed end of the artery [64], resulting in continuous blood supply to the brain via the collateral vertebra–basilar plexus in cattle. An early sealing of severed carotid arteries end (as early as 21 s in cattle observed by Gregory et al. [65]) ensures blood supply for the brain for a longer time. This prolonged consciousness risks the transmission of nociceptive neuron signals to reach the brain. False aneurysms are more common at the severed cardiac end of carotid arteries during the halal slaughtering of cattle [67].

Table 1 summarizes the effect of various stunning methods on animal welfare and meat quality.

#### 3.2.1. Mechanical Stunning

Mechanical/percussive stunning is safe for the operator and economical with a minimal recurring cost. It causes a concussion within the animal’s head achieved by a penetrative captive bolt or a non-penetrative percussion stunner. Penetrative captive bolt stunning is unacceptable in halal slaughter as it causes permanent brain injury, the animal’s inability to recover fully, and death. The concussion caused by the pins of the captive bolt alters the brain function and induces immediate, irreversible unconsciousness due to loss of evoked potential [82]. It also causes skull injuries, and the impact of the bolt leads to the transfer of mechanical energy to the skull and brain tissue, leading to the destruction of brain tissue and the direct damage by penetrating the bolt, leading to sudden brain death [45]. 

While using a captive bolt gun, operator safety risks due to the potential ricochet of the bullet, proper restraining of the head of animals within easy reach of the operator, and proper positioning of a gunshot in agitated animals are some challenges [83]. In agitated animals, the operator has to come near the head for precise hitting, which may lead to hemorrhage and cracks in the skull and the release of brain tissue [83]. FAS (Food Standard Agency, London, UK), in a week-long survey (16–22 September 2013), observed the wide-scale application of captive bolt guns and the Jarvis Beef Stunner (electric stunner, induces cardiac arrest, as replacer of conventional brisket electrode) in halal slaughtering. However, these forms of irreversible stunning are not halal-compliant and remain a major concern of Muslim consumers [57].

The non-penetrative/percussive bolt stunning is accepted in the halal slaughtering of cattle and buffalo as it is reversible, the bolt does not penetrate the skull, and there is less risk of intracerebral hemorrhage. The injury caused during this process should be temporary, and the animal skull should have no permanent injury marks [84]. However, it should be applied properly, and the stunner should not penetrate or break the skull. These stunners should be regularly cleaned to remove the accumulation of silica or carbon, which otherwise reduces the power of subsequent shots.

The application of non-penetrative captive bolt stunning needs proper aiming and suitable head restraint to reduce the mis-stun or failure of stun leading to more than one stunning attempt, thus, risk of skull and brain tissue injury and spread of brain tissue in blood circulation. Grandin [60] proposed a mushroom head with a larger diameter rather than a mushroom with a smaller head for improving stunning efficacy. Non-penetrative stunners are less effective in cattle with more hairs and also in adult cattle [85]. Finnie et al. [86] reported skull fractures and focal and diffuse injuries in some lambs stunned by non-penetrative bolt stunners and penetrative bolt stunners.

Upon applying high mechanical energy/force, the non-penetrative captive bolt stunning may fracture the skull. It also depends upon the age of animals and skull structure, bone density, and mineralization. This leads to a wide margin of error in non-penetrative stunning, which could cause a major animal welfare issue due to ineffective stun. The non-penetrative stunner that causes skull fracture is regarded as more efficient than stunning performed at a comparatively lower force or not causing skull fracture [87]. However, severe head injuries with subarachnoid hemorrhage in the adjacent brain tissue may be caused by non-penetrative stunning such as by applying a heavy mushroom head against the comparatively thin frontal bone that makes up the roof of the skull. Any damage to the skull after mechanical stunning is considered non-compliant with halal production by several HCBs. 

However, a non-penetrating mechanical stunner, although it does not penetrate the brain, has the same electroencephalogram (EEG) pattern as a penetrative captive bolt stunner and induces reversible unconsciousness, but is discouraged in cattle slaughtering due to its efficacy and animal welfare concerns [42]. The EU regulations (EC No. 1099/2009) also prohibited the use of non-penetrative captive bolt stunning for animals weighing less than 10 kg. Anil et al. [88] reported an increased risk of spread of hematogenous tissue (tissue spread through blood circulation) from the central nervous tissue under pneumatically or cartridge-operated penetrating captive bolt stunning as compared to non-penetrative captive bolt and electric stunning in cattle and sheep. Further, applying the same bolt inserted in brain tissue may spread contamination to other animals or other equipment during slaughter [60,61]. The present design of these penetrative captive bolts has been improved to check the spread of BSE (bovine spongiform encephalopathy) prions to the bloodstream but has not eliminated the risk [89,90]. All these factors caused a renewed interest in using non-penetrative captive bolt stunning. 

Zulkifli et al. [91] emphasized the importance of proper restraint and position for effective stunning. Based on electroencephalogram (EEG) readings during the slaughter of heifers and steers by penetrative captive bolt and non-penetrative low-power and high-power mechanical percussive stunning by using mushroom-headed humane killer stunning, they concluded that if stunning was performed appropriately, actual death during slaughter (assessed by cessation of heartbeat and brain death) was caused by throat cut and exsanguination. However, the penetrative stunning application was noted as the most reliable stunning method, ensuring insensibility and minimizing pain. The blood variables did not differ significantly among various groups, but penetrative captive bolt pistol-stunned animals had dramatically elevated adrenocorticotropic hormone (ACTH). The authors suggested a need to conduct further research to derive a definite conclusion.

Several Halal Certification Bodies (HCBs) have accepted non-penetrative captive bolt stunning in halal slaughter and expressed their concerns for penetrative captive bolt stunning due to irreversible unconsciousness and fear of death prior to halal throat cut. However, there are strict legal requirements placed in some countries for religious slaughter; for example, in Sweden, some halal meat producers use captive bolt stunning and electric stunning that results in the death of animals as a legal requirement of stunning for religious slaughter [19]. Heart beating has a positive effect on blood loss in animals. Any delay in bleeding after stunning may cause the heart to stop beating and prevent proper blood drainage [54], and an efficient captive bolt stunning may stop the heart from beating instantaneously [90].

The application of mechanical stunning in commercial poultry production of high line speed of slaughter (up to 10,000 birds per hour) is rather impossible, of which, if adopted, the following points are followed as prescribed under halal slaughtering of poultry such as recitation of Tasmiyyah on every bird by a Muslim or a person of the Book, and severance of prominent blood vessels in the neck region. Some views of Islamic scholars and Muslim organizations on the application of mechanical stunning in poultry are presented in Table 2. 

There have been many debates going on about halal compliance of mechanical stunning, but still, there have been a lot of doubts and concerns mostly for penetrating captive bolt stunning. As Islam advocates avoiding doubtful things, some researchers even suggest avoiding the use of captive bolt mechanical stunning for halal meat slaughter [54] and require further research to fulfill the criteria of halal slaughter [92].

#### 3.2.2. Electrical Stunning

Under electrical stunning, animals become unconscious due to the mass depolarization of neurons caused by the flow of electric current to the brain. Depending upon the frequency, site where the electrodes are placed, and strength of the electric current, it can be reversible or irreversible. Head-only electrical stunning is considered by a majority of Muslims as humane, safe, and halal-compliant. The other process of electrical stunning, viz., head-to-back, head-to-forelegs, or split current, causes the heart to stop beating leading to death and is thus not accepted in halal slaughtering of animals and birds. 

Head-only electrical stunning induces unconsciousness without any autonomous movements or responses. If not stuck, the animal regains consciousness within 20–40 min without any sign of pain or aversion [68,83]. The initiation phase of head-only electrical stunning (epileptic seizures) is regarded as painless, and a strong synergistic effect has been noticed upon halal throat cut, reducing the chances of regaining consciousness and also ensuring the animal remains alive at the time of the throat cut [93]. Due to the possibilities of wide variations in live animal weight, lean-to-fat ratio, dryness, skin thickness, and body coverings of hairs/wools, it is recommended to exsanguinate within 15 s [37,60]. Zivotofsky and Strous [94] observed the similarity between electric stunning in animals and human electroconvulsive therapy (ECT) used to treat intractable depression in humans rather than epilepsy, thus could have potential welfare issues as an electrical shock in humans is considered as a form of torture. The authors also observed gross violation of animal welfare principles and suffering in animals re-stunned after missing proper insensibility in the first stun. In sheep, there have been reports of high incidences of ineffective electrical stunning, mainly attributed to the improper positioning of electrodes [2]. 

In the application of head-only electric stunning, the duration of insensibility remains very short in cattle, and there are chances that animals can regain consciousness during exsanguination as even before death occurs due to blood loss [95]. In calves, the vertebral circulation is maintained for up to 3 min after the throat cut [96], and the time of collapse in cattle could be as long as 265 s [33]. During slaughtering, the tonic phase in cattle ranges from 2–21 s, 30–102 s clonic phase followed by the return of rhythmic breathing within 31–90 s [97]. 

The application of head-only electrical stunning also causes higher incidences of hemorrhages and broken bones in chickens [98], carcass bruising, blood splash, blood speckles, petechiae, ecchymosis, hemostasis, and bone fractures in stunned lambs [42] and adverse effect on meat color, and shear force value in cattle [70]. The significant variations in electrical resistance among animals due to size, composition, skull bone composition and size, skin thickness, dryness, hair, etc., also increase incidences of mis-stun, resulting in gross violation of animal welfare principles [99].

The strength of the electric current should be properly monitored/audited by a competent Islamic authority or HCBs regularly. To ensure the proper flow of electric current to almost the whole brain, animals should be restrained properly, and electrodes should be placed at the right place with voltage enough to overcome the total electric resistance between the electrodes. It is also recommended to check the electric stunner on dummy loads or resistors prior to using live animals to assess its efficacy [100]. From an animal welfare and halal perspective, the electric current should be enough to cause reversible insensibility for a short time, of which immediate exsanguination, proper design and maintenance of the stunner, training of workers, and proper monitoring are crucial [50].

Reversible head-only electrical stunning does not cause cardiac arrest, ventricular fibrillation, or dysrhythmias and induces unconsciousness through brain function disruption, thus making this head-only electric stunning widely acceptable for the halal slaughtering of animals [57]. In addition, the electro-immobilization and thoracic sticks are debatable in halal slaughter and animal welfare due to their ability to mask the inappropriate pre-slaughter stunning [101] potentially. Further, the thoracic stick is not considered by some halal meat importer countries as equivalent to a horizontal neck cut [83].

##### High-Frequency Head-Only Electrical Stunning

Increasing the frequency of electric current leads to decreasing the duration of each pulse and signal wavelength. The increased frequency and high voltage cause cell damage and stop the action potential initiation. Further, the increasing pulse leads to a more frequent release of action potential, leading to attaining exhaustion point (earlier for muscle and cardiac muscle than nerves), so it cannot meet the demands of the external stimulus [102], thus initiating epilepsy without muscular or fibrillation contractions. As compared to high peak force and full contraction in muscles during low-frequency electrical stunning, high-frequency electrical stunning (>1000 Hz) produces lower peak force and lower contraction of muscles against one another, thereby reducing the amount and severity of carcass damage [102]. 

The application of high-frequency head-to-body electrical stunning (1000 to 2000 Hz, square waveform) with the same amount of electric current has been reported to improve animal welfare, does not cause heart failure or death of animals, causes no convulsion associated with muscle activity, and ensures operators’ safety [103,104]. This method (head-to-body) is more reliable for animal welfare as it causes minimal or no pain to animals [51,105]. Simmons et al. [103] also reported that heart functioning in head-to-back high-frequency electric stunning (1500 and 3000 Hz) could induce spinal inhibition of the seizure movement without ventricular fibrillation. In poultry, high-frequency electric stunning (110 mA, 1500 Hz sinusoid AC) minimized seizures [106].

##### Single-Pulse Ultra-High Current (SPUC)

It is a new system of head-only electrical stunning. This technology can induce reversible unconsciousness among animals. Robins et al. [107] used applied SPUC produced by a capacitance current spike of 5000 V and 70 A for head-only stunning in cattle. The high voltage resulted in the formation of pores in the neuronal membrane (electroporation) under the influence of a high voltage gradient. The cattle stunned by this technology do not have tonic/clonic seizures, improving operator safety and meat quality; thus, this system may be approved in halal slaughter due to its reversibility [107]. Fuseini [108] successfully developed a SPUC stunner capable of generating sufficient electric voltage and current to incur neuronal electroporation. Further research work is carried out by the authors to avoid arcing and to provide additional protection for the high-voltage switch.

The recommended stunning methods in halal slaughter are presented in Table 3. 

## 4. Novel Stunning Technologies

Since the last 30 years, there has not been a major breakthrough in stunning technologies or commercialization of a novel stunning technology. Most of the research work was focused on the refinement of stunning equipment and methods already used for commercial operations. 

There is an ever-increasing interest in the potential application of electromagnetic energy to induce reversible stunning in animals, thus making this technology suitable for halal slaughter [109]. The application of electromagnetic energy to the brain increases the temperature of brain tissue, leading to hyperthermic fainting/syncope.

### 4.1. Transcranial Magnetic Stimulation

This is based on applying a changing magnetic field to the brain by placing a transcranial magnetic stimulation (TMS) probe containing a copper coil near the skull, leading to insensibility due to the generation of electric impulses within the brain of the animal. Anil and Butler [110] proposed it to induce unconsciousness in animals. An exposure of 35 or 50 Hz electromagnetic field for 5 s on 20 broilers has demonstrated its potential in inducing insensibility for 15–20 s based on EEG (predominance of theta and delta waves), tonic, and clonic reflex, muscle flaccidity, and recovery period [111]. There is a need to improve the structure and design of copper coils, so as to be accurately placed on the skull and optimize the power supply, consequently producing unconsciousness for a longer duration, making it useful in slaughterhouses. The placement of TMS for a longer duration may cause anxiety, pain, distress, or suffering in conscious animals [101]. It can potentially develop into a short-lasting reversible stunning method in animals and is still in the experimental phase. There is a need to conduct laboratory animal trials under semi-commercial conditions to establish the commercial potential and prospects for this technique [112].

### 4.2. Microwave-Induced Insensibility

Microwave irradiation is a common method of euthanasia of laboratory rats and mice by increasing the temperature of the brain (75–90 °C upon 2.5 kW for 0.68 s, 85 °C upon 10 kW for 1.25 s, 90 °C upon 10 kW at 2450 MHz for 900 ms in rats and 2450 MHz for less than 1 s in chicken) [113,114]. The high temperature inactivates the enzymes present in brain tissue, resulting in brain death, and upon controlling the irradiation in such a way that it stops normal brain function due to hyperthermia but does not harm brain tissue (a temperature below 50 °C), the insensibility can be reversible [95]. However, to be applicable to humane slaughter and halal compliance, the duration of insensibility should be sufficient to let an animal die due to blood loss upon exsanguination. The thermal tolerance of nervous tissue lies between 40–60 min at 42 °C and 10–30 min at 43 °C and thus has a sufficient time margin to ensure death due to blood loss [33,115]. For increasing the temperature of the sheep brain to 8 °C for effective stunning, 31.9 kW at 2600 MHz or 61.6 kW at 3350 MHz power was required (Rankin, 1986 personal communication to [95]). 

However, the lack of a high-power microwave generator with a targeted energy delivery and lack of shielding, occupational hazards, and scale-up of the technology are some issues that warrant immediate research on this aspect. 

### 4.3. Diathermic Syncope (DTS)

Diathermic syncope^®^ (DTS) is a novel system recently used for rendering animals insensible prior to slaughter by increasing the temperature of brain tissue by applying microwave energy [116]. This technology has been reported to provide promising outcomes during its trials in 234 heavy bull and Brahman-type cattle in Australia. This technology (by using a 20 W power setting, 160–200 kJ energy delivery) induces reversible insensibility for sufficient time (unresponsive to stimuli up to 4 min post-DTS) for a halal neck cut without any sign of distress, pain, or vocalization, does not require restraining the animal during exsanguination; EEG data with high-amplitude-low-frequency (HALF) showed an epileptic state with no visible damage to brain tissue, ensuring efficient and fast bleeding out [116]. A lower power setting (18 W) resulted in a longer time for becoming unconscious, and a higher power (25 W) setting causes overheating at skin surfaces. The authors observed that corneal reflexes were absent for 100 s and EEG suppression was noted for 80–240 s. The authors suggested that animals receiving lower energy applications may have reversible unconsciousness. On application of DTS, animals have rapid blinking and flickering of the third eyelid, with nystagmus, absence of vocalization, and slow and deep rhythmic breathing (a sign of interruption of medullary function). On completion of the DTS application, eyes became fixed and staring with the absence of eyelid movements, and absence of pupillary response to light and rapid twitching of ears were evident, as well as an absence of withdrawal response to nose prick [116]. 

However, there is a need to optimize the energy delivery to the brain as poor contact between the waveguide and the animal head may lead to energy leakage into the Faraday cage, triggering auto-cut-off switches and terminating energy application. This poor contact could be due to uneven head shape, the curvature of the head leading, movement of animals during energy application leading to energy leakage, and automatic energy termination [116]. Thus, there is a need for further research in designing a suitable delivery apparatus to ensure consistent energy delivery to the brain and proper optimization of critical energy and power parameters essential to induce a recoverable insensibility in a wide range of animals.

## 5. Post-Cut Stunning

Under post-cut stunning, animals are slaughtered as per the conventional slaughter except for applying stunning within 5 s post-cut [57]. Post-cut head-only electrical stunning is advocated as a potential halal-compliant slaughter method with improved animal welfare, absence of risk of animal death before sticking, and improved meat quality. This is followed in some European countries and is considered an improvement over animal slaughtering without stunning in ensuring animal welfare principles [66,83,117]. Post-cut head-only electrical stunning also does not cause blood splash in the carcass [42]. 

Lambooij et al. [118,119] observed that while rotating the restrainer, the animal welfare aspects of calves during slaughtering are grossly violated. They recommended a post-cut captive bolt stunning to improve animal welfare in rotating restrainers. Thus, post-cut stunning on cattle allows easier handling in an upright restraint pen prior to slaughter, and it is advised to apply efficient post-cut stunning immediately after the cut (SAWA Swedish Animal Welfare Agency, [120]), preferably within 5 s after neck cut without any further modification in between cut and stunning application [121]. 

In New Zealand, electrical immobilization is used in addition to electrical stunning of halal red meat production. The whole process comprises: (i) head-only electrical stunning (alternate current, 1.5–2.5 A, 400 V for 2–4 s) by keeping electrodes behind the ears and nose tip, (ii) making a halal cut within 10–15 s after stunning, (iii) electro-immobilization by passing a 80–90 V direct current, 10–15 pulses/s for 15–30 s by placing electrodes between the nose and the anus, (iv) clipping of esophagus and rodding of trachea followed by regular dressing [54].

## 6. Stunning: No Significant Impact on Bleeding

Proper bleed-out of the carcass is critical during halal slaughtering as blood consumption is haram in Islam (Qur’an 2: 173, 5: 3, 6: 145, 16: 115); thus, proper bleed-out of the carcass is preferred by Islamic scholars. Besides blood loss, a reference is made in the Holy Qur’an regarding the pouring or flowing of blood (Quran 6: 145). It suggests that residual blood in carcasses is not focused [36,42]. Thus, several Halal Certification Bodies (HCBs) have insisted on sufficient time for the flowing of blood from the carcass before processing [84]. Many Muslim consumers believe that stunning obstructs the blood flow and causes a reduction in expelling blood from the carcass [40,50]. Aghwan et al. [122] emphasized the proper bleeding of animals according to halal principles for improving animal welfare status, reducing pain and distress, and improving the quality and wholesomeness of meat. The removal of harmful microbes, public health protection, and extending the shelf life of carcasses may be reasons that blood removal from carcasses was emphasized in the era of lack of any refrigeration equipment [35,104].

Several published scientific findings show the non-significant effect of stunning (pre-stun or post-stun) on the bled-out/total blood loss during slaughter. Anil et al. [123] observed the bleed out of halal slaughtered sheep after electric stunning, captive bolt stunning, or slaughter without stunning and reported comparable total blood loss (blood loss % live weight) and packed cell volume (PCV). A similar finding was reported in cattle by Anil et al. [51] and Gomes Neves et al. [124]. Anil et al. [51] evaluated the bleeding efficiency in cattle by captive bolt stunning and neck cut to traditional halal slaughter without stunning and reported comparable blood loss variables, packed cell volume, and meat quality among both groups. 

Farouk et al. [42] and Masri [36] emphasized the importance of residual blood over blood loss during halal slaughter. Khalid et al. [57] compared the bleed-out of lambs in commercially used V-restraints slaughtered by traditional religious slaughter, head-only electric stunning, and post-stun head-only electric stun for 4 min bleeding under upright orientation and vertical hanging. The authors recorded a significant (*p* < 0.01) increased blood loss in the first 1 min in stunned lambs compared to traditional religious halal slaughtered lamb. However, all groups have comparable blood loss measured after 1.5 min. Although orientation changing from an upright to a vertical hanging has aided the blood loss; at the end of the bleeding process (2 min in lambs), the authors did not observe any significant variations. 

Velarde et al. [68] did not notice any remarkable differences in live weights, carcass weight at 45 min and 24 h post mortem, blood loss, and chilling losses between head-only electric stunned lambs and un-stunned lambs. The blood volume loss to body weight (4.6% and 4.3%, respectively) and killing-out (50.47% and 49.52%, respectively) were significantly higher in stunned lambs as compared to non-stunned lambs, which could be due to catecholamine secretion under stunning stress leading to peripheral vasoconstriction. Contrary to the above findings, Nakyinsige et al. [73] observed higher blood loss in traditional halal-slaughtered New Zealand white rabbits without stunning with lower hemoglobin content in m. longissimus thoracis et lumborum as compared to gas-stun-killed rabbits. Farouk et al. [42] attributed it to species-specific and slaughtering method variations. 

## 7. Thoracic Stick: Halal Compliance Way for Rapid Blood Loss

At the time of the slaughtering of sheep and cattle in head-only electric stunning in New Zealand and Australia, immediately after exsanguination, an incision is made with a knife through the thoracic inlet directed towards the heart in order to severe the brachiocephalic trunk (knife severs the brachiocephalic trunk in cattle and punctures heart in small animals) for faster bleed out and to reduce bleeding space [83]. 

Figure 2 depicts the thoracic stick cut in cattle.

The thoracic stick in sheep is crucial as all the blood supply to the brain is exclusively from carotid arteries. In cattle, vertebral arteries arising from the brachiocephalic trunk before the carotid arteries maintain sufficient blood flow to the brain to maintain its basic function. Further ballooning of carotid arteries due to spasms at the cut site made by a blunt knife helps maintain systolic pressure. Due to these factors, cattle could remain conscious for up to 2 min after slaughter [96,126]. As unconsciousness in cattle during electric stun may last up to 30–40 s, the thoracic stick hastens the bleed-out, causing a very rapid loss of blood pressure and death of an animal [127].

At present, the thoracic stick is performed after the 30 s of halal cut; until then, a sufficient amount of blood is lost and the animal is technically dead, and the animal is insensitive to pain. In the case of occlusion of the carotid arteries, the thoracic stick is performed well before it regains consciousness in electric stunning (30–40 s), and the animal remains insensitive to pain for about 5 min [128]. 

There are some concerns about its blood loss at the non-recommended site and even a second slaughter after the first halal cut. However, similar methods have been already approved (*Nahr*) and in practice during the slaughtering of camels and giraffes by stabbing in the throat followed by severing the upper part of the chest [129] and halal slaughter in Indonesia [66]. 

The Malaysian Fatwa Council, in a special seminar on the issue of thoracic sticking during halal slaughter, concluded that this should not be considered as the main halal cut and approved for halal slaughter if the proper halal cut is applied (severing the main blood vessels, gullet, and windpipe), performed at least 30 s after the halal cut, and the halal cut is the main cause of death with a thoracic stick only to aid blood loss [130]. 

## 8. Assessment of Death Prior to Halal Cut

By applying head-only electrical stunning and non-penetrative mechanical stunning, the risk of death of animals before the throat cut and exsanguination (crucial for the application of these stunning methods in halal slaughtering) is eliminated. However, in the case of the use of electrical water bath stunning in poultry, pre-stun shock, and inversion and shackling of birds, the variations in size/live weight, impedance/resistance to flow current, and lack of proper control on the sufficient electric current flow through the head of each bird pauses animal welfare, product quality, and halal compliance issues. Birds with high electrical resistance are likely to receive a lower electric current than desired, whereas birds with low resistance would receive a higher electric current than optimum, which may cause cardiac arrest and death before the throat cut and exsanguination, thus not complying with the prescribed guidelines for halal slaughtering of poultry. Hence, these birds should be immediately removed from the processing line [45]. Similarly, halal compliance concerns regarding the death of birds before slaughter have also been raised in the case of the use of gaseous stunning used for halal poultry meat production practiced in the UK, the Netherlands, and Germany as the gaseous mixture used for stunning induces death rather than unconsciousness [131]. 

The absence of muscular movements and reflexes does not clearly confirm whether animals are dead as similar spontaneous reflexes are present in both live and brain-dead animals. In modern high-throughput and at high processing speed, it is relatively impossible and impracticable to individually assess the death status of animals and remove such animals from the processing line despite the presence of halal checkers. Thus, alternatively, there is a need to further research to validate the optimum electric parameters that cause unconsciousness but not death during the stunning process. Even after brain death (irreversible cessation of cerebral activity due to loss of brain stem functions), certain reflexes associated with the spinal cord, HPA (hypothalamic–pituitary–adrenal), and thermoregulation may be present and take a longer time (ranging from several minutes to hours) for cessation of these reflexes [26,132]. Terlouw et al. [133] noted frequent reflexes like movements of the leg and neck after stunning in unconscious animals originated in the brain stem or spinal cord. These movements were observed even after 3 min from the start of bleeding, with these movements (paddling and neck reflexes) still present even after severance of the spinal cord [133]. The authors proposed an interaction mechanism among shot placement, post-stun movements, and bleeding efficiency. Jain and DeGeorgia [134] proposed the term “brain-dead reflexes” to denote such movements attributed to stimulus-provoked movements with the term “brain-dead associated automatisms” for spontaneous movements. 

The presence of a beating heart does not confirm that the animal is alive. After brain-stem death, the heart continues to beat until the end of the oxygen supply due to blood loss or circulatory arrest. The heart can beat for some time under the state of hypoxia. Some HCBs consider the presence or absence of a heartbeat as a sign of aliveness or death of animals [29]. Under exceptional situations, heart beating can be present even after a few weeks in a ventilated heart. Thus, in humans, the term heart-beating cadaver (HBC) is used for such humans. However, Jerlstrom [135] suggested that the definition of death in animal welfare contexts should be based both on the loss of brain and cardiac function. 

The bleed-out of animals also provides some insights into the living status of animals, and experienced Muslim slaughtermen can have a reliable assessment based on this parameter. The bleeding efficiency is affected by the patency and size of the sticking wound, bleeding time, blood vessel severed, carcass orientation, cardiac arrest, muscle contraction, and dressing procedures [106]. The inability of a carcass to bleed out is thus accepted as a sign of death by various HCBs such as JAKIM as outlined in [84] and Halal Food Standards Alliance of America (www.hfsaa.org, accessed on 12 September 2023). However, scientific findings suggest that bleed-out at exsanguination cannot be used as the sole criterion for assessing the live status of an animal [45]. The bleed-out does not depend on the pumping action of the heart, and some scientific reports described the bleed-out of the carcass after cardiac arrest/ventricular fibrillation [45,51,136]. 

## 9. Recommendations

To ensure the principle of animal welfare as enshrined in Islam, the training of slaughtermen and other personnel involved in animal handling, proper maintenance of slaughtering equipment, especially restraints, knife, and stunner, proper placing/aiming the stunner with reasonable force/current, and a rapid halal cut are very crucial. Further, the high workload, tedious job, risk of accidents, and lack of proper infrastructure make the training for the slaughterers crucial [137]. Continuous work pressure leads to compassion discomfort, followed by compassion stress, which finally results in compassion fatigue, resulting in poor work quality [138]. 

There is a need to conduct comprehensive research on halal slaughter regarding proper compliance of the stunning methods to clear any doubts over the stunning in halal slaughtering of animals and establish the scientific basis behind various practices of halal slaughter. The research on stunning compliance should be promoted in Muslim-majority countries. This will certainly harmonize religion and science. Further, this harmonization of religion and science should be based on aligning the religious requirements with the available scientific data to the maximum possible extent.

Furthermore, there is also a need for more scientific research in developing and modifying the present stunning technologies in the broad ambit of halal slaughtering of animals. The lack of research and the fear of hiding scientific facts during slaughtering are some factors that create doubts among Muslim consumers, resulting in their disapproval of stunning. 

There is an urgent need for undertaking extensive and in-depth research works on the commonly used stunning methods for cattle in developing countries such as non-penetrative and penetrative captive bolt stunning. The issue of reversibility of non-penetrative mechanical stunning and its impact on the brain tissue and skull bones leading to injury, which may be a permanent one, must be undertaken on a priority basis.

The application of gas stunning could be an option in halal slaughter. There is a need to conduct further scientific research on the suitability of gas stunning for halal slaughter. For halal compliance, the animal must be alive and in a reversible phase of insensibility during gas stunning, which proves a major challenge in high throughout poultry processing plants. 

The lack of proper funding opportunities for competent scientists/researchers remains the main obstacle in undertaking comprehensive and in-depth research work on various aspects of halal slaughtering. As most of the halal slaughtering research work was carried out in the developed countries to gain business opportunities in the growing halal food market, the research work carried out by Muslim scientists and their recommendations could be more acceptable by the common Muslim consumers.

## 10. Conclusions

Muslim scholars are not unanimous on the issue of the application of stunning in the halal slaughtering of animals as some scholars perceived it as more painful, obstructing free blood flow, and causing the death of animals prior to the halal cut. Several Halal Certification Bodies and Muslim-majority countries have approved reversible stunning (mainly head-only electrical stunning and non-penetrative mechanical stunning) in halal meat production. Scientific findings suggest that halal compliance stunning technologies are reversible, do not kill animals prior to the halal cut, and do not obstruct blood loss. There is a need to conduct further research on the aspects of proper identification of the death status of animals, assurance of animal welfare, proper restraints, and their proper compliance in commercial halal meat production. It is time to refine the current stunning technologies further to make these technologies fully compatible to religious principles while keeping pace with modernization at the same time.

## Figures and Tables

**Figure 1 animals-13-03061-f001:**
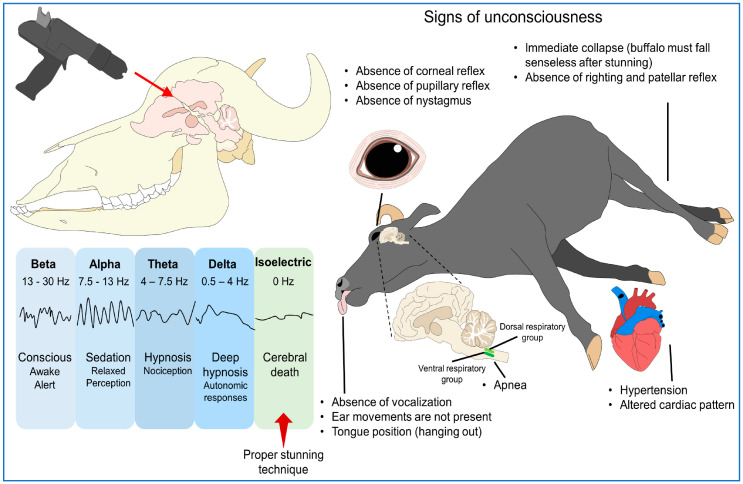
Sign to evaluate unconsciousness in buffalo (adopted from Grandin et al. [11]).

**Figure 2 animals-13-03061-f002:**
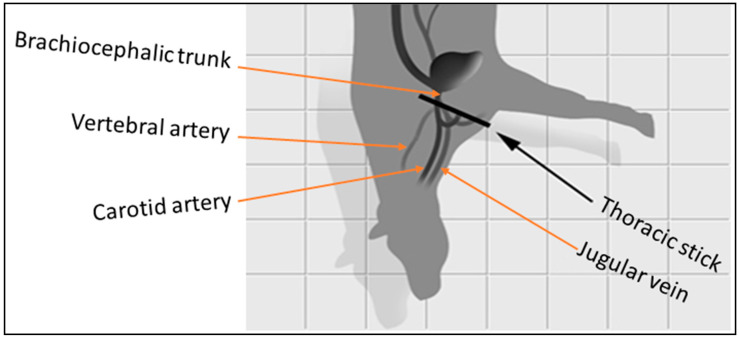
Position of thoracic stick cut in cattle (adopted and modified from Humane Slaughter Association [125]).

**Table 1 animals-13-03061-t001:** Effect of stunning methods on animal welfare and meat quality.

Animals	Stunning	Salient Findings	Reference
Calves	HOE stunning, slaughter without stunning	Stunned calves become permanently insensible immediately Time to insensibility to animals without stunning—10 s Un-stunned calves showed little or no reaction during throat cut	[47]
Lamb	HOE stunning (3 s, 250 V) and un-stunned	Electrically stunned lambs had significantly higher (*p* < 0.05) blood volume loss relative to live body weight and killing-out Higher incidences of petechiae hemorrhage in hearts (*p* < 0.05), higher in electrically stunned lambs	[68]
Lamb	HOE stunning (110 V, 50 Hz for 5 s), gas stunning (90% CO_2_ for 90 s), and without stunning	No significant difference between WHC, drip loss, shear force, cooking loss, and among all three groups No significant difference in the aging of meat by stunning	[69]
Heifer	Low- and high-power nonpenetrative mechanical stunning, penetrative and conventional halal slaughter with post-cut penetrative stunning after 10–20 s of halal cut	Meat quality of all treatments were noted as comparable (U, P, HPNP, LPNP) Stunning has not affected pH and cooking loss	[70]
Young bull	Percussive captive bolt, electric stunning, without stunning	Higher muscle glycogen content in stunned animals than slaughtered without stunning Percussive captive bolt stunned carcass had higher sensory attributes, whereas electric stunned and un-stunned carcass did not differ significantly	[71]
Geese	Water bath stunning, overfed	Less blood loss in stunned geese High-frequency electric stunning (1200 Hz) decreased the pink/red coloration of liver lobe tips in the ganders Petechial hemorrhages on the breast muscle in both sexes in high-frequency electric stunning	[72]
New Zealand white rabbit	Gas stunning-killing, halal slaughter without stunning (HS)	Halal-slaughtered carcass had significantly higher blood loss Longissimus lumborum of HS rabbit had lower residual hemoglobin HS meat has better keeping quality	[73]
Hyla rabbit	Electric stunning, halal slaughter without stunning	Halal slaughter without stunning rabbits did not have vocalization, spasm, and movement during hanging with body remaining relaxed and floppy on hanging chain Higher blood loss and pale color carcass in halal without stunning rabbits than electric-stunned slaughtered rabbits	[74]
Fallow deer	Electric stunning-TS, captive bolt-TS, captive bolt–gash cut, Electric stunning–gash cut, captive bolt–incomplete severance	Incidences of ecchymosis (total round, loin) reduced in thoracic sticked deer carcass Blood loss not affected by stunning method Exsanguination had significant (*p* < 0.001) effect with thoracic stick	[75]
Sheep	HOE stunning, correct vs. incorrect tong placement	Short duration of stunning increased the risk of a poor stun quality Current level, stun duration, or tongs’ position have no significant effect on blood splash	[76]
Veal calves	Neck cut with and without stunning during restraining and rotating, pre-cut electric stunning, post-cut captive bolt	Unconsciousness lasted for 80 s; corneal reflex was absent after 135 ± 57 s after neck cut Rotating the restrainer compromised the animal welfare principles Post-cut captive bolt stunning and pre-cut electric stunning induce immediate unconsciousness	[77]
Broiler	Captive bolt stunning, electric water bath, shackled in a cone	Captive bolt stunning resulted in a significantly higher degree of convulsion and lower blood loss as compared to electric stunning Significantly reduced thigh muscle hemorrhage in broiler restrained in a cone as compared to broiler shackled. Comparable cooking loss	[78]
Geese	9 electrical stunning methods	The loss in liver weight on removing engorged blood vessels had significantly decreasing trends (*p* < 0.05) at 350 Hz, 70 to 90 V, and 80 to 85 mA	[79]
Geese	High-frequency electric water bath stunning	Increasing the current intensity reduced the DPPH and total-SOD in goose breast meat Geese stunning at 40 mA at 500 Hz for 10 s could alleviate stunning stress and meat lipid oxidation.	[80]
Broiler	Electric stunning	Increasing carcass defects with increasing electric stunning voltage Stunning at 53 V for 10 s maximized the bleed out A beating heart is not necessary for effective bleed out in broilers	[41]
Broiler, hen, duck	Water bath electric stunning	Adequate stunning electric current did not vary significantly between broilers, hens, and ducks At high frequency electric stunning resulted in higher current flow	[81]

HOE—head-only stunning, WHC—water holding capacity, HPNP—high-power non-penetrating (0.25 caliber, 4 grain cartridge), LPNP—low-power non-penetrating (0.25 caliber, 3 grain cartridge), U—unstunned with post-cut penetrative stunning after 10–20 s of halal cut, P—penetrative (0.22 caliber, 4.5 grain cartridge), HS—halal slaughter without stunning, TS—thoracic stick, DPPH—diphenylpicrylhydrazyl, SOD—superoxide dismutase.

**Table 2 animals-13-03061-t002:** Islamic perspective on the application of mechanical stunning in halal slaughter of poultry.

Scholar/Organization	Observations
Ebrahim Desai	Birds conveyed to a single fixed blade controlled by a Muslim is not halal, and those conveyed to several fixed blades, each controlled by a Muslim, is halal
Mufti Khalid Saifullah Rahmani	During mechanical slaughtering by machine, only first birds during the slaughter of which *Tasmiyyah* was recited is halal, and the rest are haram
Halal Consultations Limited (Solihull, UK)	“All the certifier of Halal has to do for mechanized killing is ensure that the bird is not decapitated (or dead), the words of *Tasmiyyah* are recited, as required, and animal welfare rules are adhered to”
Mufti Muhammad ibn Adam al- Kawthari	Mechanical slaughter must cut two jugular veins, trachea, and esophagus, with a sharp blade with the recitation of *Tasmiyyah*, which must be recited by a man of the Book
GMWA Food Guide	During slaughter, if *Tasmiyyah* is recited by a third party, not by the slaughterman, the meat is haram
Board of Scholars (Halal Food Authority, London, UK)	Considered mechanically slaughtered poultry as halal, “In our view, the static conventional instrument of slaughter has now been transformed into a dynamic mechanical knife that facilitates mass production without compromising Halal standards”.

Source: [20].

**Table 3 animals-13-03061-t003:** Recommended stunning methods in halal slaughter.

Stunning Method	Animals
Non-penetrative captive bolt	Cattle, steers, heifer, sheep, broilers, and rabbit
Head-only electrical	Cattle, steers, heifer, sheep, broilers, rabbits, and ostrich
Gas stunning	Turkey, chicken, halal birds
Electrical water bath	Turkey, chicken, halal birds

Source: [50].

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
