# Peer review of "Stunning Compliance in Halal Slaughter: A Review of Current Scientific Knowledge"

_animals, 2023, doi:10.3390/ani13193061_

Round 1
Reviewer 1 Report
The paper presents an overview of the present scientific issues related to stunning and Halal slaughter.
Abstract: A conclusion is missing.
1. L84-91: The period of conciousness during bleeding is also a concern. More over, incorrect handling (including restraint and cut).
2.1. L154: The definition of death in Halal slaughtering is missing. Also in 8. there is not a clear description used for Halal.
5. L353: Unconsciousness, analgetic state, insensibiliy? It is unclear for the reader.
6. L430: How do you assess cessation of heart beat and brain death during slaughter.
6. L499-502: There are also other gasses than CO2 or no gas (vacuum).
References: L21: Adam is insufficient.
Author Response
Response to Reviewer 1 comments
The authors gratefully acknowledge the reviewer for critical comments and observations. These comments have helped us in improving the quality of the manuscript. We have edited the manuscript accordingly.
Further, it is certified that all the issues raised by the reviewers have been incorporated into the revised manuscript. All the changes were marked in RED color text.
A detailed description of revisions is as follows-
Comment: A conclusion is missing
Response: Conclusion in the abstract has been added.
Comment: L84-91: The period of conciousness during bleeding is also a concern. Moreover, incorrect handling (including restraint and cut)
Response: Thank you so much for the observation, we have revised the manuscript accordingly and two paragraphs on consciousness and restraints and halal cut is added in the revised manuscript.
Comment: L154: The definition of death in Halal slaughtering is missing.Also in there is not a clear description used for Halal.
Response: The definition of death has been added in the revised manuscript in section 8.
Comment: L353: Unconsciousness, analgetic state, insensibiliy? It is unclear for the reader
Response: Two separate sections on unconsciousness and consciousness state and the sentence has been edited.
Comment: L430: How do you assess cessation of heart beat and brain death during slaughter.
Response: The brain death and heart beat is checked by the isoelectic point of EEG and heart beat by stethoscope.
Comment: L499-502: There are also other gasses than CO2 or no gas(vacuum).
Response: This section has been removed from the manuscript on the recommendation by Reviewer 2.
Comment: References: L21: Adam is insufficient.
Response: The reference has been replaced with appropriate one.

Reviewer 2 Report
Halal and other religious slaughter techniques are highly interesting for both scientific and public opinion. It is known as a very controversial topic, particularly because of the potential pain that animals might perceive due to neck cutting. Proposing new alternatives to perform Halal slaughter while improving animal welfare and preserving the roots of the Halal procedure is a relevant issue for animal scientists. The present review summarizes the application of stunning methods and how this could or could not be applied following Halal requirements. While it is a highly interesting paper, it has some flaws that need to be corrected to improve the structure and presentation of the review. One issue is that the authors did not focus on a single species, making the paper very long. Another flaw is that some parts of the paper sound repetitive. For example, in several sections of the article, it is repeated that Halal slaughter does not permit skull damage. In these instances, the authors could summarize all the information regarding this specific Halal requirement and discuss it in a single section to avoid repetition. Likewise, some paragraphs need to be rewritten to improve clarity (e.g., lines 422-425). I left some particular comments hoping they can be helpful for the authors.
Simple summary. Please, add the aim of the review in this section and a brief practical application or implication according to the conclusions. For example, if stunning techniques do not seem to alter the Halal slaughtering and do not have consequences on blood flow, it could be recommended, or is it a factor that needs to be considered during the religious slaughtering of livestock?
Line 31. Consider modifying the Abstract of the article, following the Journal’s suggested order: Background: Place the question addressed in a broad context and highlight the purpose of the study; Methods: briefly describe the main methods or treatments applied; Results: summarize the article main findings; and Conclusions: indicate the main conclusions or interpretations. The aim of the study, although it seems to be included in lines 29-30, needs to be clearly stated. Likewise, include a brief conclusion of the review.
Line 32. Consider adding “religious slaughter” as a keyword.
Line 34. Consider adding an Introduction before stating the details about Halal slaughter. For example, I suggest mentioning in a couple of lines what is Halal slaughter, then including the controversial aspects of this practice, and what is the current approach that animal scientists are proposing. My suggestion is to re-order some of the current paragraphs as follows:
“1. Introduction
(Add a paragraph about what is Halal slaughter, where is practiced, who practices Halal slaughter, and the frequency of practice –maybe showing some percentages of its practice worldwide–, to give an overall background to the reader).
Normally, the slaughtering procedure is carried out by severing the main blood vessels (carotid arteries that supply oxygenated blood to the brain, and jugular veins that carry deoxygenated blood from the brain and other cells back to the heart). Death is attained due to the lack of oxygen supply to the brain [3]. The process in general would induce pain and stress to the animal, which, if not carefully monitored would compromise meat quality. The pain sensation and stress suffered by the animal would jeopardize the animal welfare aspects, which is the main concern for both the Muslim community and Western countries [4–6].
In order to overcome these issues, various stunning procedures have been developed in order to eliminate pain during slaughter. Stunning makes animals’ unconscious instantaneously and is widely used in the slaughter of animals throughout the world, even made compulsory during slaughter in some countries. With the increasing Muslim population, increase in net income, education, awareness, and meat quality assurance, the demand for Halal meat is rapidly increasing throughout the world. However, Muslim scholars are not unanimous on the application of stunning during Halal slaughter. The present review summarizes various aspects of the application of stunning technologies during the Halal slaughtering of livestock.”
2. Halal slaughter
(Include lines 35-83).
Line 42. Please, add a reference.
Lines 84-87. This topic is very interesting, and the authors could also discuss the formation of false aneurysms when conscious animals bleed without previous stunning. This is important because this might reduce blood loss and increase to time to reach unconsciousness.
Lines 87-90. Pain and its consequences (e.g., peripheral sensitization) are the main controversial topics regarding this type of slaughter. I recommend adding more information about pain perception, activation of peripheral nociceptors, and its implications on animal welfare. These articles might be helpful: https://doi.org/10.3390/ani7020011 and https://doi.org/10.3390/ani13152406.
Lines 93-95. Please, specify in which countries or regions stunning if compulsory. Also, I recommend adding that “When performed correctly, stunning makes animals unconscious instantaneously…”.
Lines 102-104. Please, add references for both perspectives (i.e., authors that state that animals should be alive during slaughtering and authors that mention that animals should be conscious during slaughter”. Additionally, please, revise the intended meaning of the sentence. I suppose that the authors meant that some scholars are in favor of stunning while others don’t. However, suggesting that stunned animals are “not alive” might be controversial and due to different interpretations. In lines 44-45 the authors mention that, for Halal slaughter, “the animal must be deemed alive”. If the meaning of this sentence is that a conscious animal is the same as an alive animal, this must be stated here.
Line 141. Please, consider discussing this article https://doi.org/10.3390/ani11041085, since improper handling and restrain are mentioned in these lines. Here, the consequences of the formation of false aneurysms could also be added in more detail than in the Introduction.
Lines 156-157. Why is it that some researchers conclude that stunning causes more pain to animals? It would be important to add some studies about it, explaining how these authors evaluated pain, which variables they used, and why they reach to this conclusion. This is essential because, during Halal slaughter, stunning is recommended to prevent or minimize pain.
Line 172. Please, add a reference.
Lines 177-182. In these lines, two ideas need to be explained. First, please, explain why the public thinks that non-stunned animals bled out better. Second, as I previously commented, when referring to an “alive animal”, explain if, according to Muslim consumers or beliefs, this is a synonym for “conscious animals”, or if this idea arises from the chance of an improper stunning technique that might cause dead before neck cutting.
Line 190: Please, delete “and birds”. These are already included in “animals”.
Line 205. More detail into signs of recovery of consciousness could be added. Please, revise these papers: https://doi.org/10.1016/j.meatsci.2016.03.011, https://doi.org/10.1016/j.meatsci.2016.03.010 and https://doi.org/10.3390/ani13152406
Line 210. Please, add a reference.
Line 221: Please, specify what the authors meant by “minimum time”.
Line 252. Please, try to link this paragraph with the next.
Lines 274-279. It would be appropriate to include details in this technique (post-cut stunning). For example, how is it applied, in which species have been applied, and why it is associated with improved animal welfare? If there are studies that compared this technique and a conventional one, it would be interesting to know if pain-related behaviors or biomarkers diminished with post-cut stunning vs. pre-cut stunning.
Lines 294-295. Could the authors discuss some studies where stunning has been shown to obstruct blood flow after neck cutting?
Table 1. Consider moving the Table just after section 2.2.
Section 4 and Section 2.2 are both called “Stunning methods in halal slaughtering”. As a recommendation, both sections could be merged into one, keeping lines 173-272 as an introductory text of Section 4, and then proceed to the different mechanical, gas, electric, etc. types of stunning.
Lines 378-387. Could the authors rewrite these lines? It is confusing to determine which methods cause reversive or irreversible consciousness because penetrative captive bolt, percussive bolt, and “mechanical stunning” are referred to in the same paragraph.
Table 2. Revise the in-text citation format for websites.
Section 6.2. Considering the aim of the article, I do not think that all the information included in this section is relevant to the present review. The information mentioned in lines 467-495 is already well known. There Is no mention of papers associated with Halal slaughter. Therefore, consider removing these lines and mentioning before the conclusions that gas stunning before Halal slaughter could be an option, but further research is needed.
Section 8. This section is very interesting, and it could easily be part of a separate paper regarding the quality of death during the Halal slaughter of livestock. However, by re-reading the title and the aim of the review, I do not consider that this section is relevant to the application of a stunning method before Halal slaughter. The subsections are sometimes very general and there is no association with Halal.
Section 9. I agree with the authors that Halal slaughter is not only controversial, but it has a wide area for research where available literature is scarce. The present review could provide evidence of the potential positive effect that stunning and other techniques that could be applied together with Halal slaughter could improve animal welfare, comply with the legal aspect of animal killing, and respect Muslim consumers and procedures.
Decision. Rejected
Author Response
Response to Reviewer 2 comments
The authors gratefully acknowledge the guidance rendered by the reviewer for critical comments and observations. These comments have helped us in improving the quality of the manuscript. We have edited the manuscript accordingly.
Further, it is certified that all the issues raised by the reviewers have been incorporated into the revised manuscript. All the changes were marked in RED color text.
A detailed description of revisions is as follows-
General comments: Halal and other religious slaughter techniques are highly interesting for both scientific and public opinion. It is known as avery controversial topic, particularly because of the potential pain that animals might perceive due to neck cutting. -----------Proposing new alternatives to perform Halal slaughter while improving animal welfare and preserving the roots of the Halal procedure is are levant issue for animal scientists. Some paragraphs need to be rewritten to improve clarity (e.g., lines422-425). I left some particular comments, hoping they can be helpful for the authors.
Response: The manuscript is thoroughly revised as per very valuable and in-depth observation by the Reviewer. The repetitive section is merged. However, the authors fully agree with the point that the manuscript covers all animal species in general rather than specific to one species in the context of Halal slaughter.
Comment: Please, add the aim of the review in this section and a brief practical application or implication according to the conclusions. For example, if stunning techniques do not seem to alter the Halal slaughtering and do not have consequences on blood flow, it could be recommended, or is it a factor that needs to be considered during the religious slaughtering of livestock?
Response: The simple summary section has been edited as per the observation.
Comment: Line 31. Consider modifying the Abstract of the article, following the Journal’s suggested order: Background: Place the question addressed in a broad context and highlight the purpose of the study; Methods: briefly describe the main methods or treatments applied; Results: summarize the article main findings; and Conclusions: indicate the main conclusions or interpretations. The aim of the study, although it seems to be included in lines 29-30, needs to be clearly stated. Likewise, include a brief conclusion of the review.
Response: The abstract has been revised as per the observations.
Comment: Consider adding “religious slaughter” as a keyword.
Response: religious slaughter added as keyword.
Comment: Line 34. Consider adding an Introduction before stating the details about Halal slaughter. For example, I suggest mentioning in a couple of lines what is Halal slaughter, then including the controversial aspects of this practice, and what is the current approach that animal scientists are proposing. My suggestion is to re-order some of the current paragraphs as follows: Introduction------The present review summarizes various aspects of the application of stunning technologies during the Halal slaughtering of livestock.” Halal slaughter---.
Response: The introduction section has been added.
Comment: Line 42. Please, add a reference.
Response: Reference has been added.
Comment: Lines 84-87. This topic is very interesting, and the authors could also discuss the formation of false aneurysms when conscious animals bleed without previous stunning. This is important because this might reduce blood loss and increase to time to reach unconsciousness.
Response: Thank you so much for your valuable observation. We have added the information on consciousness/unconsciousness. The formation of false aneurysm has been discussed in detail in L326-336 in the revised manuscript.
Comment: Lines 87-90. Pain and its consequences (e.g., peripheral sensitization) are the main controversial topics regarding this type of slaughter. I recommend adding more information about pain perception, activation of peripheral nociceptors, and its implications on animal welfare. These articles might be helpful: https://doi.org/10.3390/ani7020011 and https://doi.org/10.3390/ani13152406.
Response: We have added the desired information on pain and sensitization at L51-69. Authors are very grateful for suggesting the high-quality papers which help us the revision of this section.
Comment: Lines 93-95. Please, specify in which countries or regions stunning if compulsory. Also, I recommend adding that “When performed correctly, stunning makes animals unconscious instantaneously…”.
Response: The information about the countries pertaining to stunning in Halal slaughter has been added as desired from L 277-284 in the revised manuscript.
Comment: Lines 102-104. Please, add references for both perspectives (i.e., authors that state that animals should be alive during slaughtering and authors that mention that animals should be conscious during slaughter”. Additionally, please, revise the intended meaning of the sentence. I suppose that the authors meant that some scholars are in favor of stunning while others don’t. However, suggesting that stunned animals are “not alive” might be controversial and due to different interpretations. In lines 44-45 the authors mention that, for Halal slaughter, “the animal must be deemed alive”. If the meaning of this sentence is that a conscious animal is the same as an alive animal, this must be stated here.
Response: The reference have been added. We agree with the valuable observation made by the reviewer on the live and deemed-live status debate in Halal slaughter. However, Authors are sorry to say that most of this aspect is reported by Survey/ personal communications and thus we have included combined references for both aspects.
Comment: Line 141. Please, consider discussing this articlehttps://doi.org/10.3390/ani11041085, since improper handling and restrain are mentioned in these lines. Here, the consequences of the formation of false aneurysms could also be added in more detail than in the Introduction.
Response: The section has been revised accordingly with a focus on the restraints and false aneurysm.
Comment: Lines 156-157. Why is it that some researchers conclude that stunning causes more pain to animals? It would be important to add some studies about it, explaining how these authors evaluated pain, which variables they used, and why they reach to this conclusion. This is essential because, during Halal slaughter, stunning is recommended to prevent or minimize pain.
Response: Authors agree with the observation by the reviewer. We have edited the statements as this perception in the scholars could be due to miss-stun.
Comment: Line 172. Please, add a reference.
Response: Thank you so much for the observation. As this points on meat quality issue is taken from Fuseini et al. 2016. In the later section of manuscript, the issue of improper stunning on meat quality and carcass defects are added.
Comment: Lines 177-182. In these lines, two ideas need to be explained. First, please, explain why the public thinks that non-stunned animals bled out better. Second, as I previously commented, when referring to an “alive animal”, explain if, according to Muslim consumers or beliefs, this is a synonym for “conscious animals”, or if this idea arises from the chance of an improper stunning technique that might cause dead before neck cutting.
Response: The reference added for the perception regarding lower blood loss and live status to stunning.
Comment: Line 190: Please, delete “and birds”. These are already included in “animals”.
Response: deleted
Comment: Line 205. More detail into signs of recovery of consciousness could be added. Please, revise these papers:https://doi.org/10.1016/j.meatsci.2016.03.011,https://doi.org/10.1016/j.meatsci.2016.03.010 andhttps://doi.org/10.3390/ani13152406
Response: A section of consciousness/unconsciousness has been added. The authors are very grateful for suggesting the high-quality papers that help us revise this section.
Comment: Line 210. Please, add a reference.
Response: Reference Riaz et al., 2021 added.
Comment: Line 221: Please, specify what the authors meant by “minimum time”.
Response: The sentence has been edited with the following sentencing detailing bout the time gap.
Comment: Line 252. Please, try to link this paragraph with the next.
Response: We have revised the text accordingly.
Comment: Lines 274-279. It would be appropriate to include details in this technique (post-cut stunning). For example, how is it applied, in which species have been applied, and why it is associated with improved animal welfare? If there are studies that compared this technique and a conventional one, it would be interesting to know if pain-related behaviors or biomarkers diminished with post-cut stunning vs. pre-cut stunning.
Response: Details about post-cut stunning and its application has been added. Authors appreciated the observation but could not get any published studies that compared pain-related behavior or biomarkers.
Comment: Lines 294-295. Could the authors discuss some studies where stunning has been shown to obstruct blood flow after neck cutting?
Response: Authors appreciated the observation. In the section 6, effect of stunning on bleeding was discussed.
Comment: Table 1. Consider moving the Table just after section 2.2.
Response: Edited
Comment: Section 4 and Section 2.2 are both called “Stunning methods in halal slaughtering”. As a recommendation, both sections could be merged into one, keeping lines 173-272 as an introductory text of Section 4, and then proceed to the different mechanical, gas, electric, etc., types of stunning.
Response: Edited as per valuable suggestion.
Comment: Lines 378-387. Could the authors rewrite these lines? It is confusing to determine which methods cause reversive or irreversible consciousness because penetrative captive bolt, percussive bolt, and “mechanical stunning” are referred to in the same paragraph.
Response: The section on the mechanical stunning is thoroughly revised as per suggestion.
Comment: Table 2. Revise the in-text citation format for websites.
Response: The table formatting is edited and source of the Table from which it was modified is mentioned.
Comment: Section 6.2. Considering the aim of the article, I do not think that all the information included in this section is relevant to the present review. The information mentioned in lines 467-495 is already well known. There Is no mention of papers associated with Halal slaughter. Therefore, consider removing these lines and mentioning before the conclusions that gas stunning before Halal slaughter could be an option, but further research is needed’
Response: As suggested the gas stunning section is deleted with a brief paragraph added in the conclusion about their future prospects in the Halal slaughter.
Comment: Section 8. This section is very interesting, and it could easily be part of a separate paper regarding the quality of death during the Halal slaughter of livestock. However, by re-reading the title and the aim of the review, I do not consider that this section is relevant to the application of a stunning method before Halal slaughter. The subsections are sometimes very general and there is no association with Halal.
Response: We appreciated the observation by the reviewer. We have concised this section and keep it in the revised manuscript in very briefly keeping in mind the importance given to dead status of animals in Halal slaughter.
Comment: Section 9. I agree with the authors that Halal slaughter is not only controversial, but it has a wide area for research where available literature is scarce. The present review could provide evidence of the potential positive effect that stunning and other techniques that could be applied together with Halal slaughter could improve animal welfare, comply with the legal aspect of animal killing, and respect Muslim consumers and procedures.
Response: Thank you so much for your observation. We really appreciate it.

Reviewer 3 Report
In “Stunning Compliance in Halal Slaughter: A Review of Current Scientific Knowledge,” the authors address the issue of stunning animals before slaughtering them according to the Halal religious rules. The purpose of this review is to explore the available scientific literature on stunning methods that can be accepted in Halal meat production while ensuring animal welfare.
The authors begin by stating that the use of stunning in Halal slaughtering of animals is a means to reduce stress, alleviate pain and minimize fear, but it is still questioned by a part of the Muslim scholars, mostly because they have doubts about the efficacy of the current methods and their compatibility with religious principles. Thus, they recommend further research into the efficacy of the current methods, with particular reference to their capacity to induce unconsciousness rather than death of the animals, which is unacceptable in Halal slaughter.
The strength of the paper is that it covers an issue – the stunning of animals during ritual slaughtering – that is still controversial and will benefit from a review of existing literature as well as providing a focus on Halal slaughter and additional recommendations.
However, I believe that the work suffers from a development of the theme, which should be enhanced in order to make it clearer. For example, in the subparagraph devoted to “Mechanical stunning”, the discussion about penetrative and non-penetrative captive bolts is rather confusing, ranging from “The percussive bolt stunning is accepted in Halal slaughtering as it is reversible, the bolt does not penetrate the skull, and there is less risk of intracerebral hemorrhage.” (lines 378-380), to “The concussion caused by the pins of the captive bolt alters the brain function and induces immediate, irreversible unconsciousness” (Line 382-383), and again “Captive bolt stunning results in irreversible stunning due to brain death” (Line 422). In the middle, there is information referring to penetrative and non-penetrative captive bolts and their effects, as well as their acceptability or unacceptability, which is poorly linked and mixed with other hints to information about sanitary concerns, technical issues, effects on animal welfare, normative requirements, etc. Furthermore, no clear indication of the animal species to which the described technique can be used is provided.
I believe that the various parts (method, animal species, effects, animal welfare issues, regulatory issues, religious issues, for example) should be better organized, to make the review more suitable to its goals.
This is a general remark that I believe applies to almost each subparagraph.
I have some more specific suggestions that I will explain below.
- Paragraph 2 starts with an interesting description of the various perspectives and interpretations of Shariah law. The final sentence (Lines 118-119) should be better linked with the previous remark, for example, by adding “However”, to emphasize that not all Muslim scholars “have started approving stunning …” (Line 116-117).
- The subparagraph “Opponent of stunning” should be arranged better. I suggest beginning with the citation form Fuseini et al. and then displaying the main concerns, distinguishing scientific ones (such as animal welfare issues) from religious ones. The comments about which concerns have been addressed should follow. In this respect, the information in Lines 128-129 should be moved forward.
- Line 125, please state whether or not the Humane Slaughter Act is in force in the United States.
- Lines 144-148, the statement is not clear, and it sounds out of place.
- Line 149, the species of animals were never mentioned previously. It should be pointed out that the citation about birds is related to the remarks that follow.
- I propose reorganizing the subparagraph “Stunning methods in halal slaughtering” as the previous one, by beginning with the citation from Nakyisinge et al. and displaying the main concerns related to some of the points listed.
- In Paragraph 4, it is not clear how the effect of the orientation of animals on the bleed-out is consistent with the effect of stunning (Line 310-311). Similarly, in Lines 319-321, it is unclear with respect to which other method any significant variations were observed.
- Line 363, I’m not sure why a Table summarizing the effects of various stunning methods was included in the paragraph devoted to Thoracic stick. By the way, I observe that a Table summarizing the stunning methods allowed by the laws in force in the countries cited in the paper should be added.
- Table 1, please clarify what “Experimental design” implies and check the column “Salient findings,” because some effects associated with slaughtering without stunning are listed. Furthermore, I was perplexed as to why the word “Recommendation” had been included.
- Paragraph 6, the species to which the various stunning methods can be applied should be firstly specified in each subparagraph.
- Lines 398-406, please check how these concepts fit with the rest of the subparagraph.
- Lines 401-406, the concepts expressed in these sentences appear to be unrelated to the discussion as well as to themselves. Please, check and clarify.
- From Lines 422-424, the subparagraph sounds repetitive and confusing. The concepts should be reorganized and stated more clearly. I had trouble differentiating between penetrating and non-penetrating captive bolt impacts, as well as understanding the type of stunning they cause. Line 433 contains a reference to the blood variables that is neither well contextualized nor explained. Successively, there are several pieces of information that sound like “separated hints”: from the FAS survey revealing the use of captive bolt guns, to the comment that it is not Halal compliant (and thus?), to the issue over delays between stunning and sticking, to a few remarks about the operator safety risk when using a captive bolt gun, to a few examples of problems affecting diverse species …
- Table 2: Is only information on the Islamic perspective on the use of mechanical stunning in Halal slaughter of poultry available? If not, please add further information or explain why you chose poultry.
- Paragraph 7: Please clarify which species are likely to benefit from the novel stunning technologies.
- Paragraph 8, Lines 673-675: do you mean that HCBs can decide in different ways? On which basis? Can you please include a bibliography?
- Subparagraph 82, Lines 680-681: The absence of movements contrasts with spontaneous reflexes, which entail movement. Please clarify already from this point what “absence of movements” means in this context. In this subparagraph, a brief description of what irreversible stunning and irreversible mechanic stunning may cause is abruptly followed by a reference to undue delays in stunning and Halal slaughter, after which a consideration about the fact that “the presence of a beating heart does not confirm that the animal is alive” is stated as a consequence (“Thus”). Anyway, the concepts, in my opinion, are unrelated. Moreover, there are no remarks on Muslim scholars’ or HCBs’ positions.
- Figure 2: In the Note, any comment on the presence of pain (although not confirmed) is made.
- Subparagraph 8.3: please add some information regarding the bleeding efficiency parameter and a final note, after outlining the concerns.
- Paragraph 9. The recommendation to enforce the training of slaughtermen and the other personnel involved is remarkable. Anyway, this issue is poorly addressed in the manuscript.
- Lines 745-748: this statement should be thoroughly justified. I understand that the authors think that research conducted by Muslim scientists in a Muslim-majority country may be suitable to gather trust in the results from Muslim people. However, science should remain “neutral” and take advantage of every serious contribution. On the other hand, harmonization between religion and science should be sought through aligning religious requirements with scientific data and knowledge, where this is possible. Otherwise, one might think that a harmonization is “found” or “not found” without the sufficient neutrality.
- Lines 190, 510, 747 and 752: please check “animals and birds”: birds are animals.
- Line 752: The lack of transparency is a major issue that impacts all types of slaughter. This topic is not covered in the paper. Please either explain it more properly or remove it.
- Lines, 762-765, see the comment above referred to the same concept expressed in Lines 745-748.
Author Response
Response to Reviewer 3 comments
The authors gratefully acknowledge the guidance rendered by the reviewer for critical comments and observations. These comments have helped us in improving the quality of the manuscript. We have edited the manuscript accordingly.
Further, it is certified that all the issues raised by the reviewers have been incorporated into the revised manuscript. All the changes were marked in RED color text.
A detailed description of revisions is as follows-
Comment: In “Stunning Compliance in Halal Slaughter: A Review of Current Scientific Knowledge,”----------------The strength of the paper is that it covers an issue – the stunning of animals during ritual slaughtering – that is still controversial and will benefit from a review of existing literature as well as providing a focus on Halal slaughter and additional recommendations.
Response: Thank you so much for your valuable observations and positive comments.
Comment: However, I believe that --------------- can be used is provided. I believe that the various parts (method, animal species, effects, animal welfare issues, regulatory issues, religious issues, for example) should be better organized, to make the review more suitable to its goals. This is a general remark that I believe applies to almost each subparagraph.
Response: Authors really appreciate the valuable suggestion. We have edited the whole manuscript and reorganized various section as per the suggestions.
Comment: Paragraph 2 starts with an interesting description of the various perspectives and interpretations of Shariah law. The final sentence (Lines 118-119) should be better linked with the previous remark, for example, by adding “However”, to emphasize that not all Muslim scholars “have started approving stunning …” (Line 116-117).
Response: Edited as per suggestion
Comment: The subparagraph “Opponent of stunning” should be arranged better. I suggest beginning with the citation form Fuseini et al. and then displaying the main concerns, distinguishing scientific ones (such as animal welfare issues) from religious ones. The comments about which concerns have been addressed should follow. In this respect, the information in Lines 128-129 should be moved forward.
Response: The section arranged as per the suggestions.
Comment: Line 125, please state whether or not the Humane Slaughter Act is in force in the United States.
Response: Added as in force in the USA.
Comment: Lines 144-148, the statement is not clear, and it sounds out of place.
Response: we have edited as per suggestion.
Comment: Line 149, the species of animals were never mentioned previously. It should be pointed out that the citation about birds is related to the remarks that follow.
Response: Edited and birds added
Comment: I propose reorganizing the subparagraph “Stunning methods in halal slaughtering” as the previous one, by beginning with the citation from Nakyisinge et al. and displaying the main concerns related to some of the points listed.
Response: The section is organized as per the observation.
Comment: In Paragraph 4, it is not clear how the effect of the orientation of animals on the bleed-out is consistent with the effect of stunning (Line 310-311). Similarly, in Lines 319-321, it is unclear with respect to which other method any significant variations were observed.
Response: The section edited as per observation.
Comment: Line 363, I’m not sure why a Table summarizing the effects of various stunning methods was included in the paragraph devoted to Thoracic stick. By the way, I observe that a Table summarizing the stunning methods allowed by the laws in force in the countries cited in the paper should be added.
Response: The Table position is shifted after stunning section as per suggestion of reviewer 2 to make it more relevant.
Comment: Table 1, please clarify what “Experimental design” implies and check the column “Salient findings,” because some effects associated with slaughtering without stunning are listed. Furthermore, I was perplexed as to why the word “Recommendation” had been included.
Response: In Table 1: the information related to without stunning is deleted and the only Animals, Stunning, salient finding and references have been added.
Comment: Paragraph 6, the species to which the various stunning methods can be applied should be firstly specified in each subparagraph.
Response: Thank you so much for your valuable suggestions. We have added Table 3 to summarize this information.
Comment: Lines 398-406, please check how these concepts fit with the rest of the subparagraph.
Response: The section has been edited.
Comment: Lines 401-406, the concepts expressed in these sentences appear to be unrelated to the discussion as well as to themselves. Please, check and clarify
Response: The lines deleted
Comment: From Lines 422-424, the subparagraph sounds repetitive and confusing. The concepts should be organized and stated more clearly. I had trouble differentiating between penetrating and non-penetrating captive bolt impacts, as well as understanding the type of stunning they cause. Line 433 contains a reference to the blood variables that is neither well contextualized nor explained. Successively, there are several pieces of information that sound like “separated hints”: from the FAS survey revealing the use of captive bolt guns, to the comment that it is not Halal compliant (and thus?), to the issue over delays between stunning and sticking, to a few remarks about the operator safety risk when using a captive bolt gun, to a few examples of problems affecting diverse species …
Response: This section on Mechanical stunning is thoroughly revised as per suggestions.
Comment: Table 2: Is only information on the Islamic perspective on the use of mechanical stunning in Halal slaughter of poultry available? If not, please add further information or explain why you chose poultry
Response: As in the poultry slaughter, due to high degree of mechanization and slaughter rate, the issue of mechanical stunning in the Halal slaughter become more relevant. Further, the published information on this aspects for the other animals is still lacking.
Comment: Paragraph 7: Please clarify which species are likelyto benefit from the novel stunning technologies.
Response: As these technologies are yet to commercialized, so its applicability and the species of animals to be benefitted in future is not clear yet. However, the experiments were undertaken in poultry and cattle.
Comment: Paragraph 8, Lines 673-675: do you mean that HCBs can decide in different ways? On which basis?Can you please include a bibliography?
Response: This section is concised based on the observation by reviewer 2 and hence these sentences were deleted.
Comment: Subparagraph 82, Lines 680-681: The absence of movements contrasts with spontaneous reflexes, which entail movement. Please clarify already from this point what “absence of movements” means in this context. In this subparagraph, a brief description of what irreversible stunning and irreversible mechanic stunning may causeis abruptly followed by a reference to undue delays in stunning and Halal slaughter, after which a consideration about the fact that “the presence of a beating heart does not confirm that the animal is alive” is stated as a consequence (“Thus”). Anyway, the concepts, in my opinion, are unrelated. Moreover, there are no remarks on Muslim scholars’ or HCBs’ positions
Response: Edited as per suggestion
Comment: Figure 2: In the Note, any comment on the presence of pain (although not confirmed) is made.
Response: The fig 2 is deleted during concising this section.
Comment: Subparagraph 8.3: please add some information regarding the bleeding efficiency parameter and a final note, after outlining the concerns.
Response: Added as per suggestions
Comment: Paragraph 9. The recommendation to enforce the training of slaughtermen and the other personnel involved is remarkable. Anyway, this issue is poorly addressed in the manuscript
Response: The information about the importance of training added in the manuscript.
Comment: Lines 745-748: this statement should be thoroughly justified. I understand that the authors think that research conducted by Muslim scientists in a Muslim-majority country may be suitable to gather trust in the results from Muslim people. However, science should remain “neutral” and take advantage of every serious contribution. On the other hand, harmonization between religion and science should be sought through aligning religious requirements with scientific data and knowledge, where this is possible. Otherwise, one might think that a harmonization is “found” or “not found” without the sufficient neutrality.
Response: Thank you so much for your observation; we have edited the sentence to maintain the neutrality of science.
Comment: Lines 190, 510, 747 and 752: please check“ animals and birds”: birds are animals.
Response: Bird deleted.
Comment: Line 752: The lack of transparency is a major issue that impacts all types of slaughter. This topic is not covered in the paper. Please either explain it more properly or remove it.
Response: The controversial sentence deleted.
Comment: Lines, 762-765, see the comment above referred to the same concept expressed in Lines 745-748.
Response: Thank you so much for your observation. We have edited the sentences.

Round 2
Reviewer 2 Report
It is deeply disconcerting to observe that the authors of the article have employed a figure from another publication (https://doi.org/10.3390/ani13152406) and incorporated it as Figure 1 within their manuscript. Such conduct is not only ethically dubious but also constitutes a violation of copyright law, as the use of figures and images without the authors' explicit consent is prohibited. It is imperative to investigate whether a similar situation arises concerning Figure 2. Merely citing the source does not suffice. It is incumbent upon the authors to remove any non-original figures not created by themselves from their manuscript.
Such actions should warrant immediate rejection of an article.
Author Response
Dear Reviewer,
Fig 1 is adopted from Grandin et al. review paper published in Animals under open access, and the source is duly cited.
Fig 2, we have replaced the previous figure with this one adopted and modified from the Humane Slaughter Association under section captive bolt piston. On its website,(https://www.hsa.org.uk/publications/online-guides) it is mentioned that
"The HSA online guide e-learning series has been developed to enable access, free of charge, to the HSA’s education and training resources anywhere in the world."
The HSA is a charitable organization promoting the humane treatment of all food animals.
Reviewer 3 Report
I observed that you appropriately improved the paper. Thank you.
Author Response
Thank you for your kind evaluation of our paper.